# Contextuality without nonlocality in a superconducting quantum system

Markus Jerger[1], Yarema Reshitnyk[2], Markus Oppliger[3], Anton Potočnik[3], Mintu Mondal[3], Andreas Wallraff[3], Kenneth Goodenough[4], Stephanie Wehner[4], Kristinn Juliusson[5], Nathan K. Langford[4,6] & Arkady Fedorov[1,2]

Classical realism demands that system properties exist independently of whether they are measured, while noncontextuality demands that the results of measurements do not depend on what other measurements are performed in conjunction with them. The Bell–Kochen–Specker theorem states that noncontextual realism cannot reproduce the measurement statistics of a single three-level quantum system (qutrit). Noncontextual realistic models may thus be tested using a single qutrit without relying on the notion of quantum entanglement in contrast to Bell inequality tests. It is challenging to refute such models experimentally, since imperfections may introduce loopholes that enable a realist interpretation. Here we use a superconducting qutrit with deterministic, binary-outcome readouts to violate a noncontextuality inequality while addressing the detection, individual-existence and compatibility loopholes. This evidence of state-dependent contextuality also demonstrates the fitness of superconducting quantum circuits for fault-tolerant quantum computation in surface-code architectures, currently the most promising route to scalable quantum computing.

[1] ARC Centre of Excellence for Engineered Quantum Systems, The University of Queensland, St Lucia, Queensland 4072, Australia. [2] School of Mathematics and Physics, The University of Queensland, St Lucia, Queensland 4072, Australia. [3] Department of Physics, ETH Zurich, CH-8093 Zurich, Switzerland. [4] QuTech, Delft University of Technology, Lorentzweg 1, 2611 CJ Delft, The Netherlands. [5] Quantronics group, SPEC, CEA, CNRS, Université Paris-Saclay, CEA Saclay, Gif-sur-Yvette 91191, France. [6] Kavli Institute of Nanoscience, Delft University of Technology, P.O. Box 5046, 2600 GA Delft, The Netherlands. Correspondence and requests for materials should be addressed to A.F. (email: a.fedorov@uq.edu.au).

Realistic models of nature aim to describe the predictions of quantum mechanics using underlying hidden variables (HVs), which determine the properties of the system ahead of time. The best known examples are local HV theories, which seek to explain the predictions of quantum entanglement under the combined assumptions of realism and locality[1]. The divide between quantum and classical physics, however, runs deeper than the feature of entanglement. The Bell–Kochen–Specker theorem[2,3] considers nonconextual HV models that are defined without reference to locality. The Bell–Kochen–Specker theorem shows that, already for qutrit systems, it is not possible to define such a model that is consistent with quantum theory.

While the original theorem is difficult to test, the discovery of noncontextuality inequalities[4–7] makes tests of noncontextual models accessible experimentally even in the presence of imperfections. Noncontextuality tests have been carried out in a range of different physical systems and dimensionalities, including neutrons[8], trapped ions[9,10], single photons[11–13] and spins of nitrogen-vacancy centres in diamond[14,15], but all of these experimental tests introduced additional loopholes. As in tests of local realism, insufficient detector efficiencies lead to the detection loophole. Here ignoring undetected events introduces a selection bias that can be exploited to find a consistent HV explanation. The individual-existence and compatibility loopholes are important for noncontextuality tests, which require the comparison of multiple outcomes in a measurement context[3,4,16]. If measurements are performed jointly[17], it is not always possible to establish a meaningful operational definition of an individual measurement. It is therefore critical to obtain individual measurement outcomes for each measurement, for example, by making measurements sequentially[18]. The compatibility loophole arises when imperfections cause sequential measurements to be imperfectly commuting[9,19,20]. The compatibility loophole has been addressed in both photonic[12] and trapped-ion[9] experiments, but the detection and individual-existence loopholes have only been addressed using high-efficiency readout in trapped ions[9,10]. The three loopholes have only been addressed simultaneously in a two-qubit scenario[9], where it remains possible to construct explanations involving quantum entanglement.

In this experiment, we realize the Klyachko–Can–Binicioğlu–Shumovsky (KCBS) state-dependent noncontextuality test[4] with a tunable superconducting qutrit. By engineering deterministic, binary-outcome readouts, we violate a noncontextuality inequality while addressing the detection, individual-existence and compatibility loopholes in a singe experiment without entanglement. This evidence of state-dependent contextuality in superconducting quantum circuits demonstrates their suitability for fault-tolerant quantum computation using magic state distillation[21].

## Results

### The Klyachko–Can–Binicioğlu–Shumovsky test

The KCBS state-dependent noncontextuality test[4] uses five different observables $A_i$ ($i = 1, 2, ..., 5$) with binary outcomes $\pm 1$. The test involves measuring the five pairs of observables, called measurement contexts, $\{A_1, A_2\}$, $\{A_2, A_3\}$, $\{A_3, A_4\}$, $\{A_4, A_5\}$ and $\{A_5, A_1\}$, chosen such that each observable is measured in two different contexts. Noncontextual HV models predict that the total observable correlations for outcome pairs are bounded by[4]

$$\langle A_1 A_2 \rangle + \langle A_2 A_3 \rangle + \langle A_3 A_4 \rangle + \langle A_4 A_5 \rangle + \langle A_5 A_1 \rangle \geq -3. \quad (1)$$

This inequality can be violated in quantum mechanics. Here we consider a qutrit system, with five dichotomic observables $A_i = 2|l_i\rangle \langle l_i| - 1$ corresponding to states represented by vertices of the pentagram shown in Fig. 1. Each observable can be

described by a pair of projectors $\{|l_i\rangle \langle l_i|, I - |l_i\rangle \langle l_i|\}$ associated with outcomes $\{+1, -1\}$. The states connected by edges of the pentagram are orthogonal, ensuring that the corresponding observables, $A_i$ and $A_{i+1}$, (and their measurement operators) commute, making them compatible observables. Quantum mechanics predicts that the left side of (1) evaluates to $5 - 4\sqrt{5} \simeq -3.944$ for a qutrit in the ground state, $|0\rangle$. This is the maximum quantum violation of inequality (1) (ref. 6).

**Superconducting qutrit.** We encode a qutrit into a transmon-type multilevel quantum circuit[22] incorporated into a three-dimensional microwave copper cavity (Fig. 2a,b). The three lowest-energy eigenstates of the weakly anharmonic transmon form the qutrit's logical states, with allowed transition frequencies of $v_{01}^{max} = 6.939$ GHz between the ground and first excited states, and $v_{12}^{max} = 6.623$ GHz between the first and second excited states, corresponding to an anharmonicity of $\alpha \equiv v_{12} - v_{01} = -314$ MHz. The qutrit is dispersively coupled with strength $g = 17.9$ MHz to the cavity's fundamental mode (with bare frequency $v_c = 7.3014$ GHz and linewidth 2.4 MHz). We detect the state of our transmon qutrit in the usual way via the state-dependent frequency shift of the cavity, by measuring the amplitude and phase of a probe signal transmitted through the cavity. This signal is then amplified by a Josephson parametric amplifier[23], a cryogenic high-electron-mobility transistor amplifier and a chain of room-temperature amplifiers. The high-fidelity single-shot detection enabled by the parametric amplifier ensured that each experimental trial produced a definite outcome, thus closing the detection loophole.

**Binary-outcome readout.** In this experiment, we close the individual-existence loophole by performing efficient sequential measurements with classical, binary outcomes. Critical to this is our ability to implement partially projective dichotomic measurements on the qutrit system. For our transmon system, the state-dependent cavity frequency shift is[22]

$$s_j = -\chi_j + \chi_{j-1}, \quad \chi_{j \geq 0} \approx \frac{(j+1)g^2}{v_{j,j+1} - v_c}, \quad \chi_{-1} = 0. \quad (2)$$

Ordinarily, this gives distinguishable measurement responses for all states of the qutrit[24], resulting in a fully projective

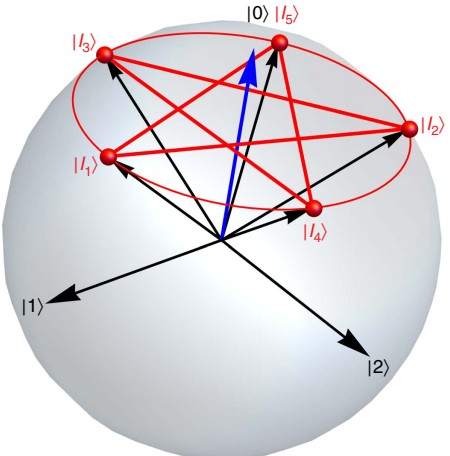

**Figure 1 | KCBS pentagram.** The qutrit eigenstates are $|i\rangle$ with $i = 0, 1$ and 2. One can construct five qutrit states $|l_i\rangle$ corresponding to five dichotomic observables $A_i = 2|l_i\rangle \langle l_i| - 1$. States connected by edges of the pentagram are orthogonal, assuming compatibility of the associated observables. Each pair of compatible measurements forms a context, and each observable is included in two different contexts. The states of the pentagram are chosen to provide maximum contradiction with noncontextual HV models.

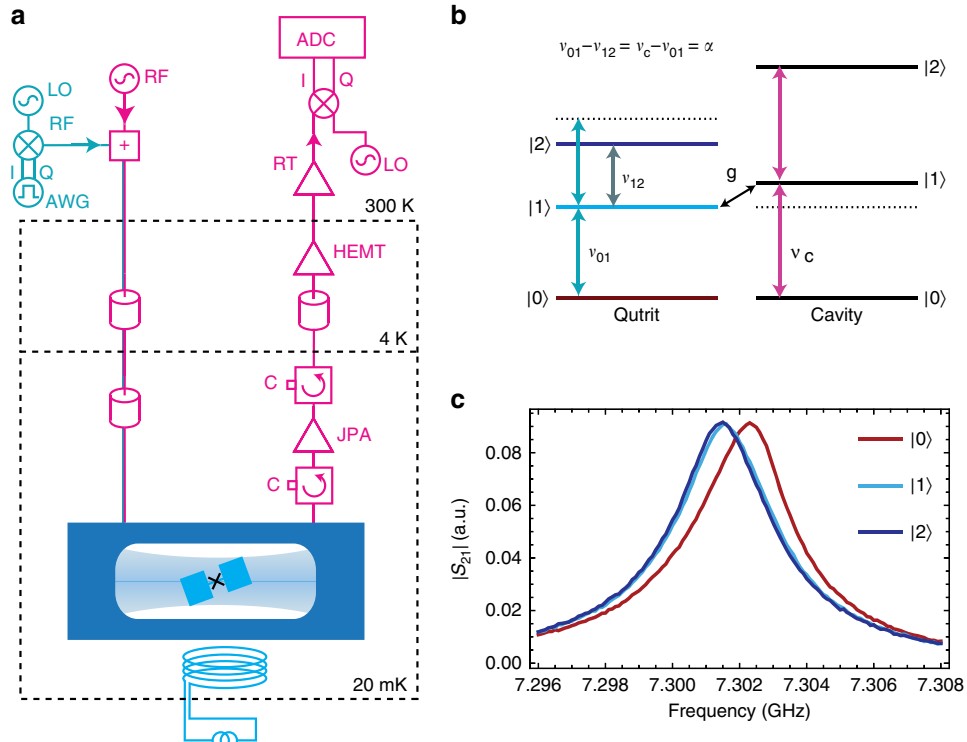

**Figure 2 | System and measurement set-up.** (**a**) Simplified diagram of the measurement set-up (see Methods for details). (**b**) The energy level diagram of a qutrit coupled to a microwave cavity when the dispersive shifts of the cavity frequency are identical for the first and second excited states of the qutrit. The scheme realizes the binary-outcome projective measurement of the qutrit on its ground state $M_{|0\rangle}$. (**c**) Transmission through the readout cavity with the qutrit in different basis states. After state preparation, a square microwave pulse with a frequency close to the resonant frequency of the cavity is applied for several microseconds. The plot indicates the normalized amplitude of measured transmitted signal integrated over 2 μs. The dispersive shifts for $|1\rangle$ and $|2\rangle$ are close to identical, not allowing the measurement to distinguish between the two states.

measurement. By choosing the qutrit detuning $v_{01} - v_c \simeq \alpha$ (Fig. 2b), we match two of the dispersive shifts, $s_1$ and $s_2$ (Fig. 2c), making the corresponding measurement responses indistinguishable. Probing the cavity therefore quickly erases coherences between the ground state and the excited states, but leaves the coherence between the first and second excited states intact, realizing a dichotomic measurement along $|0\rangle$, associated with the observable $M_{|0\rangle} = 2|0\rangle\langle 0| - 1$ with projectors $\{|0\rangle\langle 0|,$ $I - |0\rangle\langle 0|\}$. For detailed information on the effect of the readout pulse on the state of a qutrit state for different detunings, including a theoretical model, experimental verification and calibration procedures, see (ref. 25). Probing the cavity for 350 ns, we reach a single-shot contrast of $\approx 96\%$ between $|0\rangle$ and $|1\rangle$, $|2\rangle$ limited primarily by thermal excitation and decay of the qutrit state during the readouts. To generate measurements in arbitrary directions $|l_i\rangle$ from $M_{|0\rangle}$, we apply unitary rotations before and after measurement (Fig. 3a). Each measurement procedure starts by sending microwave pulses to the qutrit to rotate the desired measurement basis (defined by one of the KCBS states $|l_i\rangle$; see Fig. 3b,c) onto the ground-state readout basis. After a readout pulse and delay of 475 ns for cavity ring-down, further microwave pulses return the qutrit to its initial reference frame, necessary to allow subsequent measurements to be implemented independently.

**Testing compatibility**. Preserving coherence in the subspace orthogonal to the measurement direction is also crucial for ensuring the compatibility of context-independent sequential measurements. Since noncontextuality tests aim to falsify noncontextuality using the assumptions of noncontextual realism, which contain no notion of compatibility, it is important to ask

why test protocols only consider compatible measurements. It is well established that individual outcome probabilities for incompatible observables will depend on the order in which they are measured, but this overt contextuality does not reveal any further insight into the nature of reality. However, restricting attention to compatible measurements allows a study of whether context dependence still remains when this overt contextuality is absent. In practice, experimental imperfections make the actual measurement procedures only approximately compatible. This loophole can be addressed by an extended KCBS inequality[19]

$$\langle A_1 A_2 \rangle + \langle A_3 A_2 \rangle + \langle A_3 A_4 \rangle + \langle A_5 A_4 \rangle + \langle A_5 A_1 \rangle \geq \\ -3 - (\varepsilon_{12} + \varepsilon_{32} + \varepsilon_{34} + \varepsilon_{54} + \varepsilon_{51}). \quad (3)$$

Here the order of the observables in the two-outcome correlations $\langle A_i A_j \rangle$ corresponds to the timing order for two corresponding sequential measurements, and $\varepsilon_{ij}$ are the operational bounds for the incompatibility of these measurement procedures. A bound on incompatibility[19]:

$$\varepsilon_{ij} = \left| \langle A_j | A_j A_i \rangle - \langle A_j | A_i A_j \rangle \right|, \quad (4)$$

with $A_j$ measured before/after $A_i$ can be established separately (Supplementary Information).

**Protocol and measurements of correlations**. In the final protocol, we measure the five combinations $\langle A_1 A_2 \rangle$, $\langle A_2 A_3 \rangle$, $\langle A_3 A_4 \rangle$, $\langle A_4 A_5 \rangle$ and $\langle A_5 A_1 \rangle$, and their reverse-order variants, followed by calibration blocks to detect phase drifts of the cavity signal. As the qutrit is operated in a dilution refrigerator at 20 mK, its thermal state is close to the ground state. To avoid residual thermal population, we begin each experimental trial by

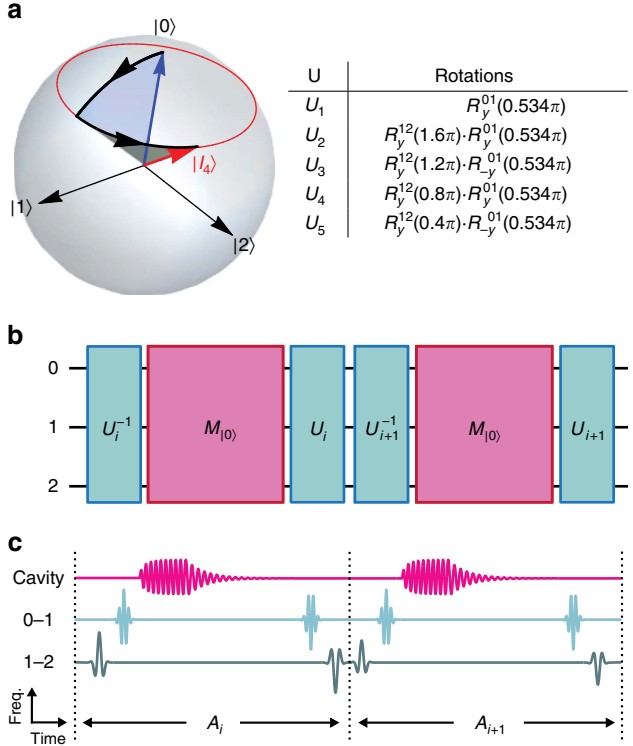

**Figure 3 | Measurement potocol. (a)** Unitary transformations of the qutrit ground state to the KCBS states. Each $U_i$ can be decomposed into one or two rotations $R_{\hat{n}}^{i,i+1}(\phi)$, where $\phi$ is a rotation of angle about the axis $\hat{n}$ in the qutrit subspace spanned by $\{|i\rangle, |i+1\rangle\}$. The rightmost pulse in a product is applied first in time. The trajectory of the state under transformation $U_4$ is shown as an example. **(b)** The measurement protocol includes two sequential projecting measurements $M_{|0\rangle}$ onto the ground state with unitary transformations before and after each measurement. The unitaries rotate the measurement axis into one of the states of the KCBS pentagram. **(c)** The actual experimental sequence for each pair of measurements. Measurement of the $M_{|0\rangle}$ observable is implemented with a cavity probe signal and the qutrit rotations $R_{\hat{n}}^{i,i+1}(\phi)$ are constructed with microwave pulses applied at the qutrit transition frequencies.

applying an initialization readout tone to the cavity to project and post-select the desired ground state, rejecting $\sim 10\%$ of all data points. A further delay of 565 ns allows the cavity to ring-down before the measurement sequence begins. The whole sequence is repeated $2^{21}$ times, triggered every 100 µs. For each observable $A_i$, the same procedure (set of pulses) was used to implement the measurement independent of measurement context, with the cavity transmission signal detected, integrated and discriminated to assign a dichotomic outcome $\pm 1$. The outcomes were recorded to the hard drive and were later used to calculate expectation values $\langle A_i \rangle$ and correlations $\langle A_i A_j \rangle$. The results used to test inequality (3) (and its reverse-order counterpart) are presented in Table 1. For all pairs, the first measurement yields expectation values very close to the ideal value of 0.105(6), with the second measurement consistently displaced due to decoherence of the qutrit during the sequence. We find a sum of correlations of $-3.489(1)$ and the threshold including incompatibility bounds of $-3.352(2)$.

**P-value calculation.** Using the standard analysis of such inequalities, we violate the KCBS noncontextuality inequality equation (3) by $> 53$ s.d. Inspired by the extended inequality derived in ref. 19, the compatibility loophole was treated by

formalizing the problem as a hypothesis test without any assumptions on compatibility and bounding incompatibility of the measurements in a separate hypothesis test[26]. The null hypothesis that the experiment is described by a noncontextual HV model with compatibility $\epsilon \lesssim 0.0413$ (see Methods) is rejected with a P-value $< 2.96 \times 10^{-575}$. A separate test of the compatibility condition rejects the hypothesis that the observables are more incompatible with a P-value $< 4.1 \times 10^{-4}$. Our analysis requires only the assumption that the devices perform the same in every single trial and the no-memory assumption without any additional assumptions on compatibility of the measurements or on the measurement contrast.

**Discussion**

Our results strongly contradict the predictions of noncontextual HV models, closing two common loopholes: the detection loophole, via high-fidelity, deterministic single-shot readout and the individual-existence loophole[18], using separate, sequential measurements. The compatibility loophole was treated by violating an extended inequality[19] and, independently, by formulating the problem in the form of a hypothesis test without any assumptions on compatibility and bounding the incompatibility of the measurements in a separate hypothesis test.

As a key ingredient in addressing these loopholes, we implemented sequential dichotomic qutrit measurements that project out one target state without disturbing the information stored in the remaining two-dimensional subspace. This allows a classical result from the first measurement to be obtained before implementing the setting to be used for the second measurement. Our results demonstrate that quantum mechanics departs from predictions of noncontextual realism, without reliance on nonlocality or entanglement correlations, and provide evidence of the contextuality resource in superconducting circuits. While we used the simpler state-dependent inequality for demonstration of contextual nature of the superconducting circuits, the state-independent test will be the straightforward extension of our experiment.

One key point that differentiates our noncontextuality analysis for an indivisible system from a similar analysis for Bell inequalities with locality constraints is that it is difficult to avoid the need for additional i.i.d. and no-memory assumptions for measurements on a single system. Since quantum contextuality can be simulated by a classical system with memory[27], these loopholes will most likely remain for any Kochen–Specker tests without nonlocality. In another case, for the finite-precision loophole[28,29], debate continues about whether this loophole can be closed in principle[30–32]. It remains an important open challenge to identify a clear, general prescription for how to implement a noncontextuality test with minimal assumptions.

**Methods**

**Sample and cavity.** The qutrit was fabricated on an intrinsic Si substrate in a single step of electron beam lithography followed by shadow evaporation of two Al layers with an oxidation step between the depositions. The design of the circuit is identical to the one in ref. 33 and consists of two sub-millimetre size capacitor plates connected via a line interrupted by a d.c. superconducting quantum interference device (SQUID), playing the role of a magnetically tunable Josephson junction. Magnetic flux supplied by a superconducting coil attached to the copper cavity is used to control the transition frequencies of the qutrit. The qutrit has maximum transition frequencies of $\nu_{01}^{max} = 6.950$ GHz between the ground and first excited states, $\nu_{12}^{max} = 6.635$ GHz between the first and second excited states, corresponding to a anharmonicity of $\alpha = 314$ MHz and charging energy of $E_C/h = 288$ MHz as shown in Fig. 2b. At the working point of the qutrit, selected by the magnetic field bias, we measured decay and coherence times of $T_{1,1} = 17.4$ µs, $T_{1,2} = 6.2$ µs, $T_{1,2 \to 1} = 18.1$ µs, $T_{1,2 \to 0} = 9.5$ µs, $T_{2,01}^* = 6.6$ µs and $T_{2,12}^* = 4.6$ µs.

The qutrit was incorporated into a three-dimensional microwave copper cavity attached to the cold stage of a dilution cryostat (Fig. 2a). The cavity was coupled asymmetrically to the input and output microwave ports with corresponding external quality factors of $Q_{in} \simeq 80,000$ and $Q_{out} = 4,200$ for transmission

### Table 1 | Violation of the KCBS inequality.

| (i,j) | $\langle A_i A_j \rangle$ | | $\langle A_i \rangle$ | $\langle A_j \rangle$ | $\epsilon_{ij}$ | |
|---|---|---|---|---|---|---|
| (1,2) | − 0.6947(5) | | 0.0744(7) | 0.1475(7) | 0.073(1) | |
| (2,1) | | − 0.7009(5) | 0.0741(7) | 0.1530(7) | | 0.079(1) |
| (2,3) | | − 0.7080(5) | 0.0748(7) | 0.1470(7) | | 0.072(1) |
| (3,2) | − 0.7001(5) | | 0.0808(7) | 0.1488(7) | 0.068(1) | |
| (3,4) | − 0.6907(5) | | 0.0820(7) | 0.1551(7) | 0.073(1) | |
| (4,3) | | − 0.6996(6) | 0.0784(7) | 0.1511(7) | | 0.073(1) |
| (4,5) | | − 0.6992(5) | 0.0781(7) | 0.1500(7) | | 0.072(1) |
| (5,4) | − 0.7051(5) | | 0.0768(7) | 0.1477(7) | 0.071(1) | |
| (5,1) | − 0.6986(5) | | 0.0779(7) | 0.1452(7) | 0.067(1) | |
| (1,5) | | − 0.7052(5) | 0.0753(7) | 0.1469(7) | | 0.072(1) |
| $\sum$ | − 3.489(1) | | | | 0.352(2) | |
| $\sum$ | | − 3.513(1) | | | | 0.367(2) |

Correlations $\langle A_i A_j \rangle$ contribute to the left side of equation (3). We also provide $\langle A_i A_j \rangle$ for the equation with the reversed order of measurements. Single expectation values $\langle A_i \rangle$ and $\langle A_j \rangle$ are used to evaluate bounds $\epsilon_{ij}$ on compatibility contributing to the right side of equation (3). Inequality $\sum \langle A_i A_j \rangle \geq -3 - \sum \epsilon_{ij}$ is experimentally violated for forward and reversed orders by > 53 and 56 s.d.'s, respectively.

measurements with the internal quality factor of the cavity was measured in the separate runs as $Q \sim 10,000$ at mK temperatures.

To measure transmission, a signal from a microwave generator (RF) was applied to the input port of the cavity. Microwaves transmitted through the cavity were amplified by a Josephson parametric amplifier, high-electron-mobility transistor amplifier at 4 K and a chain of room-temperature amplifiers. The sample at 20 mK was isolated from the higher-temperature fridge stages by three circulators (C) in series. The amplified transmission signal was down-converted to an intermediate frequency of 25 MHz in an IQ mixer driven by a dedicated local oscillator (LO), and digitized by an analogue-to-digital converter for data analysis.

**Readout.** To implement single-shot readout, we used a Josephson parametric dimer amplifier[23] (JPDA) as a preamplifier of the signal. The Josephson parametric dimer amplifier consists of two coupled non-linear resonators and can be operated in the non-degenerate mode if a pump tone frequency is set between resonance frequencies of the resonators. In our experiment, the pump tone was set at 7.058 GHz providing amplification of 25 dB gain and 12.5 MHz bandwidth centred at the readout frequency $v_c$. Two circulators installed between the readout cavity and JPDA, combined with the readout cavity itself, eliminated any effect of the pump tone on the qutrit.

**Hypothesis test.** Experimental tests of HV models can be formulated as a hypothesis test, where the null hypothesis (to be rejected) is that the measurement statistics can be modelled using HVs[26]. To this end, the experiment is recast as a set of trials of a game that can be won with a maximum probability of $\beta_{win}$ if the experiment were governed by a specific noncontextual HV model. Specifically, we test an i.i.d. model (the devices behave the same in each trial) in which the compatibility of measurements obeys a guaranteed limit as in (4) (Supplementary Information). This limit is then tested separately. To this end, it is convenient to phrase the compatibility condition of (4) in terms of probabilities instead of expectation values as $|\Pr(A_j = a_j | \#1 = j) - \Pr(A_j = a_j | \#1 = i, \#2 = j)| \leq \epsilon_{(i,j)}$, where we use #1 and #2 to indicate the order in which we make the measurements $A_j$ labelled $i$ and $j$, and $a_j$ denotes the outcome of measurement $j$. An $\epsilon$-incompatible model assumes that

$$\left\| \frac{1}{5} \sum_{(i,j)} \epsilon_{(i,j)} \right\| \leq \epsilon. \qquad (5)$$

For the KCBS inequality, a trial is won if the two outcomes of a context are not equal. The total number of wins is recorded over the whole experimental run of $n$ trials. The $P$-value is then the probability that the game could have been won at least that many times given a noncontextual HV model with incompatibility $\epsilon$. In this experiment, we recorded 3,912,769 wins out of 4,603,450 trials, which implies that the $P$-value $\leq 2.96 \times 10^{-575}$. A second, parallel hypothesis test is formulated to test the incompatibility bound of (5).

**Data availability.** The measurement data that support the findings of this study are available in UQ eSpace: http://dx.doi.org/10.14264/uql.2016.207.

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

## Acknowledgements
We thank Pascal Macha, Andrés Rosario Hamann and Kirill Shulga for help at the early stage of the experiment. We also thank Fabio Costa and Clemens Müller for useful discussions. M.J. and A.F. were supported by the Australian Research Council Centre of Excellence CE110001013. Y.R. was supported by the Discovery Project DP150101033. A.F. was supported in part by the ARC Future Fellowship FT140100338. K.J. was supported by the CCQED network. K.G. and S.W. are supported by STW and an NWO VIDI grant.

## Author contributions
A.F., M.J. and N.K.L. designed the experiment. M.J. and Y.R. performed the experiment. M.O., A.P., M.M. and A.W. designed and fabricated the Josephson parametric amplifier. K.J. designed and fabricated the qutrit. M.J. carried out data analysis to calculate expectation values, correlations and s.d.'s. K.G. and S.W. formulated the results in the form of hypothesis test and calculated *P*-values. A.F., M.J., N.K.L., S.W. and K.G. wrote the manuscript. A.W. and Y.R. commented on the manuscript. A.F. supervised the project.

## Additional information

**Competing financial interests:** The authors declare no competing financial interests.

