## [Peer Review File · Nature Communications]

Reviewer #1 (Remarks to the Author):

In the manuscript an experiment on superconducting qutrits is presented that shows a violation of the KCBS-inequality, and hence the experiment cannot be explained by a noncontextual theory. Each of the correlations has been measured about 2 million times, yielding a relative error of the order 10^{-3} . The actual violation of the inequality is moderate, but still has a high overall significance is 49 standard deviations. This demonstrates an excellent quantum control over the experiment. The test of noncontextuality is realized by sequential measurements of the actual observables. It is notoriously difficult to implement sequential measurements of degenerate observables and achieving this high reliability and high repetition rate is remarkable.

The central claim of the paper is that a violation of the noncontextuality hypothesis has been demonstrated, free of the detection, individual existence, and compatibility loophole. In this sense the experiment is a repetition of Ref. 7, without having state-independence, but featuring the use of the indivisible qutrit. Note, however, that the violation of the state-independent qutrit-inequality by Yu and Oh has been demonstrated before, e.g., Phys. Rev. Lett. 109, 150401. It is a bit surprising that the experiment was not aiming for this more fundamental inequality. Summarizing, I would see the impact and importance of the current experiment not so much in testing the foundations of quantum mechanics, but rather in demonstrating very good quantum control in superconducting qutrits.

A major concern is how the incompatibility of the measurements is treated. Contrary to what the authors claim, compatibility is hardwired into the definition of contextuality, cf., e.g., the abstract of Ref. 18. Unfortunately, the experiment shows a very clear influence of the measurement of the first observable on the second observable, cf. Table 1. This dependence is even context dependent, as can, e.g., be seen when comparing the observable 1 in the sequences (2,1) and (5,1) (the significance of this is more than 10 standard deviations). Another concern is the choice of the error model, Eq. (3), which is obtained applying the third approach of Ref. 18. The assumptions that make this inequality plausible are not mentioned in the current manuscript but it seems that it is mandatory to obtain ϵ_{AB} for a great variety of different state preparations. This has not been done in the experiment and this is neither mentioned nor are the implications discussed. Other error models (Ref. 19, 29, etc.) might be better suited.

I have a couple of remarks which I collected while reading the manuscript. a) It is true that contextuality is necessary for magic-state quantum computing, but I doubt that this is related to the KCBS inequality. b) Whether quantum theory describes individual experimental runs is connected to the minimal interpretation of quantum mechanics; maybe a less controversial first sentence could be found. c) The noncontextuality experiments on NV-centers might also be worth to be mentioned. d) "We find a sum of correlations ... and incompatibility bound ..." is not very clear. Is the former Eq. (1) and the latter Eq. (3)? e) The hypothesis test seems a bit over the top given a significance of 49 standard deviations. However, the explicit theory presented in the Supp. Info. may turn out to be very useful for future experiments.

Reviewer #2 (Remarks to the Author):

The authors demonstrate contextuality without nonlocality with a superconducting qutrit system while the detection, individual-existence and compatibility loopholes are closed. Before this experiment, there is only one experiment with a two-qubit system that was able to close the three loopholes simultaneously. However, possibility concerning the involvement of quantum entanglement remains. The author constructed 5 KCBS state of with a superconducting qutrit and tested the inequality of total observable correlations by statistical measurement for five pairs via a high fidelity binary-outcome readout. The extended KCBS inequality (eq. 3) is experimentally

demonstrated by showing violations of more than 49 and 58 standard deviations for forward and reversed orders, respectively.

The manuscript is clearly written and the experimental data seems supporting their claim. However, I have some concerns about the readout of the experiment. It is well known the photons introduced by the readout pulse can induce dephasing of qubits, and such effect can influence the analysis of the outcome of the experiment. In the manuscript, the effect of readout pulse amplitude on the superconducting qutrit decoherence is missing. The high fidelity binary-outcome readout they developed is a key tool for their experiment. The authors cited their previous work (ref. 23) for the binary-outcome readout, but I only find it mentioning slightly shifts in sweet spot (on the order of a few MHz) for different readout powers. I think the readers will find such analysis helpful and interesting in agreeing with their conclusion.

Therefore I suggest to add the details of their calibration concerning the effect of the readout pulse amplitude on the superconducting qutrit dephasing in the current manuscript. Is there any difference between qutrit dephasing and qubit dephasing induced by readout photons in the cavity? Do they have a model to explain?

Reviewer #3 (Remarks to the Author):

This article presents the results of a new test of the (Bell)-Kochen-Specker theorem for individual qubits systems which succeeds in simultaneously closing several key loopholes - the detection, compatibility and individual-existence loopholes - improving on previous work in this direction that has not succeeded in closing them all. Key to this advance is the system the authors use -three-level superconducting systems - which are novel in this context and allow high-fidelity, non-destructive projection measurements to be performed in a 3-dimensional Hilbert space.

Using this system, the authors demonstrate the convincing violation of the state dependent KCBS inequality in a generalised form taking into account the imperfect compatibility of measurements. Although state-independent tests are more desirable (a point that could be mentioned in the paper), they are much harder to perform and remain a key next step in this research direction. Although recent loophole-free Bell-tests have also demonstrated quantum contextuality (enforced by the locality), the ability of contextuality to manifest itself in non-divisible systems means that this result is nonetheless important.

The statistical analysis of the results performed is convincing and in line with the normal for tests of quantum nonlocality and contextuality, and is presented nicely in the paper and in more detail in the supplementary material (modulo a few formatting errors and the ungainly notation of writing "P - Value" instead of "p-value").

One of the key difference in the analysis of noncontextuality inequalities for individual system vis à vis Bell-inequalities is the need to make the i.i.d. assumption that the authors discuss in the SM. Indeed, since quantum contextuality can be simulated with memory (and one cannot rule this out on grounds of locality for a single system), one cannot rule out such noncontextual HV theories. This point should perhaps be mentioned, and the paper "Memory Cost of Quantum Contextuality" (DOI: 10.1088/1367-2630/13/11/113011) is relevant to this point.

Thus, for single systems, one cannot hope to close all possible loopholes as one might for Bell-inequalities, since this "memory" loophole remains, in addition to the finite-precision problem (although it is debatable that this is really a loophole). Given this, the paper seems to go as far as is possible for closing loopholes for single system Kochen-Specker tests, except for performing state-independent tests, and thus represents a key advance made possible by a very clever experimental setup.

Overall, I think this paper is very suitable for publication in Nature Communications, with the minor points mentioned above taken into account. In addition, I have three very-minor, miscellaneous comments:

1) it is perhaps a stretch to call noncontextuality inequalities "recent" (paragraph 2), since they have been around for almost a decade.

2) (before Eq. (3)): I think it is a little misleading to attribute this "overt contextuality" to the Heisenberg Uncertainty Principle; it is simply due to noncommutativity of observables, and one can still have incompatible measurements with an underlying realist hidden variable theory.

3) In the discussion: the phrase "the last important loophole" is slightly misleading, since it would at face value imply that other loopholes are unimportant, which is not what is meant.

Reviewers' comments:

Reviewer #1 (Remarks to the Author):

In the manuscript an experiment on superconducting qutrits is presented that shows a violation of the KCBS-inequality, and hence the experiment cannot be explained by a noncontextual theory. Each of the correlations has been measured about 2 million times, yielding a relative error of the order 10^{-3} . The actual violation of the inequality is moderate, but still has a high overall significance is 49 standard deviations. This demonstrates an excellent quantum control over the experiment. The test of noncontextuality is realized by sequential measurements of the actual observables. It is notoriously difficult to implement sequential measurements of degenerate observables and achieving this high reliability and high repetition rate is remarkable.

Response to the reviewer: We thank Reviewer #1 for their positive comments. We are pleased that he/she recognizes the excellent quantum control of our experiment and the difficulty to realize sequential measurements of degenerate observables. We are also happy to see that he/she found the high reliability and high repetition rate of our experiment remarkable.

The central claim of the paper is that a violation of the noncontextuality hypothesis has been demonstrated, free of the detection, individual existence, and compatibility loophole. In this sense the experiment is a repetition of Ref. 7, without having state-independence, but featuring the use of the indivisible qutrit. Note, however, that the violation of the state-independent qutrit-inequality by Yu and Oh has been demonstrated before, e.g., Phys. Rev. Lett. 109, 150401. It is a bit surprising that the experiment was not aiming for this more fundamental inequality. Summarizing, I would see the impact and importance of the current experiment not so much in testing the foundations of quantum mechanics, but rather in demonstrating very good quantum control in superconducting qutrits.

Response to the reviewer: The goal of this experiment was to demonstrate a contradiction with HV theories in a three-level system under the strictest conditions of such tests so far. For this purpose, a state-independent test would have been no stronger or more conclusive than the state-dependent test. The KCBS inequality is the simplest scenario that contradicts HV theories. Thus, choosing the simplest scenario for the simplest indivisible system was, to our opinion, the clearest demonstration of the underlying concept. While the review is correct when pointing out that the KCBS inequality is not the specific test required to enable magic-state distillation, demonstrating the state-dependent contextuality is nevertheless a valuable conceptual step for superconducting circuits.

A major concern is how the incompatibility of the measurements is treated. Contrary to what the authors claim, compatibility is hardwired into the definition of contextuality, cf., e.g., the abstract of Ref. 18. Unfortunately, the experiment shows a very clear influence of the measurement of the first observable on the second observable, cf. Table 1. This dependence is even context dependent, as can, e.g., be seen when comparing the observable 1 in the sequences (2,1) and (5,1) (the significance of this is more than 10 standard deviations).

Response to the reviewer: Perfect compatibility is not possible to realize in an experiment even in principle. Thus, we were not surprised to see the influence of the measurement of the first observable on the second one (we note that this influence is predominantly due to decoherence between measurements not by the measurement procedure itself). The reviewer also noticed the context dependence of this influence. For this particular example

of sequences (2,1) and (5,1), the value of observable 1 changes by 0.007 which is about a 0.4% difference in the population of the ground state. Such a small difference can also be explained by differences in the decay of the states associated with observables 2 and 5 prior to measurement of observable 1.

It is not possible to avoid such influences in any realistic experiment and that the assumption of perfect compatibility is always compromised. Moreover, it is the very reason why we addressed the compatibility loophole in our experiment and this remark underlines the added value of our work compared to many others (including Phys. Rev. Lett. 109, 150401, mentioned by the reviewer), where the results actually relied on the assumption of the perfect compatibility.

To answer the remark more specifically, we state precisely that compatibility is hardwired in the definition of contextuality, e.g., in the formal description of the HV model we test in section 1.3 of the Supplementary Material. Condition 3 there (called bounded incompatibility), does precisely "hardwire" compatibility into the HV model. We also highlight this in the name of the HV model that we test, which is called an ϵ -bounded i.i.d. NCHV model, where the ϵ is the bound on the incompatibility.

We test this NCHV model using a null hypothesis test. The null hypothesis (to be rejected) is that the experiment is described by an ϵ -bounded i.i.d. NCHV model. The P-value is then the probability that we observed a behaviour at least as extreme as observed, maximized over all such HV models.

The objective of section 1.5 in the Supplementary Material is then to gain additional confidence about the compatibility of the observables in this particular experiment, that is, to bound the amount of incompatibility ϵ . We can, of course, only test this ϵ for our particular experimental setup, and not for all conceivable HV models that we maximize over in the definition of the P-value above.

As the referee points out, such ϵ -s can, of course, depend on the context. It is for this reason that the overall ϵ is defined as an average over the individual $\epsilon_{((i,j))}$, where $\epsilon_{((i,j))}$ is the bound on the incompatibility of the individual context (i, j). Since we assume that the model is i.i.d., considering such an averaged ϵ is sufficient.

We give a separate P-value for the test of incompatibility, and for the overall null hypothesis that we are testing (namely the null hypothesis of an ϵ -bounded i.i.d. NCHV model, which is explained in section 1.6 and 1.7 of the Supplementary Information).

Changes to manuscript:

To avoid any further such lack of clarity for the readers of our article, we have changed the main text before Eq. (4) on page 3. Instead of saying "A bound on incompatibility can be established as [18] ... ", we say "A bound on incompatibility [18] ... can be established separately (see Supplementary Information)."

Another concern is the choice of the error model, Eq. (3), which is obtained applying the third approach of Ref. 18. The assumptions that make this inequality plausible are not mentioned in the current manuscript but it seems that it is mandatory to obtain ϵ_{AB} for a great variety of different state preparations. This has not been done in the experiment and this is neither mentioned nor are the implications discussed. Other error models (Ref. 19, 29, etc.) might be better suited.

Response to the reviewer: We remark that while we have mainly referred to Ref. 18 in the main part of the text, there is an entirely self-contained analysis in the Supplementary

Material without reliance on any of assumptions of Ref. 18. The reason why we have mainly referred to Ref. 18 is because the ideas do, of course, stem in part from Ref. 18, to which we like to give credit. In the supplementary, we explicitly state a definition of the HV model.

Possibly, the referee is referring to the statements in Ref. 18 that the experimenter would need to test the incompatibility condition by “preparing” different distributions over hidden-variables, or possibly “experimentally accessible” ones. Translating this observation to the our formalism, this means (see also the remark above) that while in the P-value of the null hypothesis test we maximize over all ϵ -bounded i.i.d. NCHV models, we can, of course, only test the ones accessible in our experiment. Our description of the experimental apparatus is using classical and quantum mechanics, but there is no experimental procedure known to prepare distributions over HV for otherwise unknown HV models. Of course, we could simply realize different state preparations, but then we would be doing precisely the same test for each of them, using, for example, that the model is again i.i.d. meaning we perform the same preparation of the (in general unknown) HV model in every trial of the experiment.

To be formally precise, we have thus assumed that the HV model is i.i.d. This allows us to obtain statistical confidence on the incompatibility parameters, which is not discussed in Ref. 18.

Changes to manuscript: We thank the referee for pointing out that we may have been insufficiently clear on this point in our writing. We introduce the following changes to the manuscript:

- At the beginning of section 1.5 of the Supplementary Material we add the paragraph: *“Note that this means that we bound ϵ only for our experimental setup, and not for all conceivable HV models appearing in the maximization in the P-value definition of (1). Obtaining a bound for all ϵ is not possible, since we would need to be able to probe HV models experimentally which would require knowledge about this HV theory in order to prepare suitable probe states. We remark that for a fixed ϵ , our experiment rejects ϵ -bounded i.i.d. NCHV models with an essentially negligible P-value.”*
- To make clear that our analysis is self-sufficient and not based on Ref. 18 in “P-value calculation” on page 4, instead of saying *“This can be formalized as a hypothesis test [24].”* we now say (note that reference numbers were changed by the introduction of new references): *“Inspired by the extended inequality derived in Ref. 19, the compatibility loophole was treated by formalizing the problem as a hypothesis test without any assumptions on compatibility and bounding the incompatibility of the measurements in a separate hypothesis test [26].”*
- To state the assumptions that were used in the analysis we add the following sentence in the main text at the end of “P-value calculation” on page 4: *“Our analysis requires only the assumption of i.i.d., that is, the devices perform the same in every single trial and the no-memory assumption without any additional assumptions on compatibility of the measurements or on the measurement contrast.”*
- In “Discussion” on page 4, instead of saying *“The last important loophole, the compatibility loophole, was treated via an extended inequality [18].”* we now say *“The compatibility loophole was treated by violating an extended inequality [19] and, independently, by formulating the problem in a form as a hypothesis test without any assumptions on compatibility and bounding incompatibility of the measurements in a separate hypothesis test.”*

I have a couple of remarks which I collected while reading the manuscript. a) It is true that contextuality is necessary for magic-state quantum computing, but I doubt that this is related to the KCBS inequality.

Response to the reviewer: Ref 19 shows that the ability to perform magic-state distillation is linked to state-dependent contextuality. The referee correctly points out that this is a more complex context and thus requires a more complex test. However, the KCBS inequality is the simplest scenario that demonstrates state-dependent contextuality and can therefore be viewed as a valuable conceptual step towards the goal of demonstrating the viability of surface code error correction in superconducting circuits.

b) Whether quantum theory describes individual experimental runs is connected to the minimal interpretation of quantum mechanics; maybe a less controversial first sentence could be found.

Changes to manuscript: We changed the first sentence. Instead of saying *“Quantum theory prescribes possible measurement outcomes and outcome probabilities, but makes no predictions about individual experimental runs.”* we now say: *“Realistic models of nature aim to describe the predictions of quantum mechanics by underlying hidden variables (HVs), which determine the properties of the system ahead of time.”*

c) The noncontextuality experiments on NV-centers might also be worth to be mentioned.

Changes to manuscript: We introduced two more references Ref. 14, 15 to acknowledge experiments with NV-centers.

d) "We find a sum of correlations ... and incompatibility bound ..." is not very clear. Is the former Eq. (1) and the latter Eq. (3)?

Changes to manuscript: We changed this sentence: *“We find a sum of correlations of -3.489(1) and the threshold including incompatibility bounds of -3.336(2).”*

e) The hypothesis test seems a bit over the top given a significance of 49 standard deviations. However, the explicit theory presented in the Supp. Info. may turn out to be very useful for future experiments.

Response to the reviewer: We agree that it is more common to present the number of standard deviations above the threshold. Our manuscript is in fact the first work where a P-value evaluation for a test of contextuality was performed. We believe that this more rigorous and self-contained analysis was worthwhile and, in particular, allowed us to avoid reliance on any assumptions used in the derivation of the inequality. We totally agree that this approach will be very useful for future work on contextuality and thank our reviewer for acknowledging the potential of this approach.

Reviewer #2 (Remarks to the Author):

The authors demonstrate contextuality without nonlocality with a superconducting qutrit system while the detection, individual-existence and compatibility loopholes are closed. Before this experiment, there is only one experiment with a two-qubit system that was able to close the three loopholes simultaneously. However, possibility concerning the involvement of quantum entanglement remains. The author constructed 5 KCBS state of with a superconducting qutrit and tested the inequality of total observable correlations by statistical measurement for five pairs via a high fidelity binary-outcome readout. The extended KCBS inequality (eq. 3) is experimentally

demonstrated by showing violations of more than 49 and 58 standard deviations for forward and reversed orders, respectively.

The manuscript is clearly written and the experimental data seems supporting their claim.

Response to the reviewer: We thank reviewer #2 for their positive assessment.

However, I have some concerns about the readout of the experiment. It is well known the photons introduced by the readout pulse can induce dephasing of qubits, and such effect can influence the analysis of the outcome of the experiment. In the manuscript, the effect of readout pulse amplitude on the superconducting qutrit decoherence is missing. The high fidelity binary-outcome readout they developed is a key tool for their experiment. The authors cited their previous work (ref. 23) for the binary-outcome readout, but I only find it mentioning slightly shifts in sweet spot (on the order of a few MHz) for different readout powers. I think the readers will find such analysis helpful and interesting in agreeing with their conclusion. Therefore I suggest to add the details of their calibration concerning the effect of the readout pulse amplitude on the superconducting qutrit dephasing in the current manuscript. Is there any difference between qutrit dephasing and qubit dephasing induced by readout photons in the cavity? Do they have a model to explain?

Response to the reviewer: As rightly pointed by a reviewer effect of the readout on the superconducting qutrit is crucial for understanding of the binary-outcome measurement for a qutrit. We give this information on page 2 in subsection “Binary-outcome measurement.” More specifically, if the qutrit is tuned in the sweep spot the dephasing induced by readout photons *“quickly erases coherences between the ground state and the excited states, but leaves the coherence between the first and second excited states intact”*.

The effect of the readout photon on the qutrit was extensively studied in Ref. 23. The effect of erasing coherence between the ground state and the excited states while not affecting the coherence between the first and second excited states was experimentally proven using the Ramsey fringes, the partial and full state tomography for the qutrit as well as using the full process tomography of the measurement. Ref. 23 also contains the theoretical model and detailed calibration description.

Possibly, the referee concern originated from the statement in Ref. 23: *“Another observation is that the sweet spot slightly shifts in frequency (on the order of a few MHz) for different readout powers.”* We emphasize that this statement was related not to the shift of the qutrit frequency by the measurement photons but to the fact that in order to realize the binary outcome readout the qutrit has to be tuned to a particular frequency, called the sweet spot, which slightly (on the order of a few MHz) depends on the readout power used in the experiment. More specifically, we expect the sweet spot at $\nu_{01} - \nu_c \simeq \alpha$. However the precise detuning has to be determined for each readout power by a procedure prescribed in Ref. 23. Once the qutrit is tuned to the sweet spot with the magnetic field, the effect of the readout photons can be described as the projection of the qutrit onto its ground state.

Changes to manuscript:

In “Binary-outcome readout” section on page 2 instead of saying: “This was limited primarily by thermal excitation and decay of the qutrit state during the readouts (for more details, see Ref. [25].” we now say: “For the detailed information on the effect of the readout pulse on the state of a qutrit state for different detunings, including a theoretical model, experimental verification and calibration procedures see Ref. [25].”

Reviewer #3 (Remarks to the Author):

This article presents the results of a new test of the (Bell)-Kochen-Specker theorem for individual qubits systems which succeeds in simultaneously closing several key loopholes - the detection, compatability and individual-existence loopholes - improving on previous work in this direction that has not succeeded in closing them all. Key to this advance is the system the authors use -three-level superconducting systems - which are novel in this context and allow high-fidelity, non-destructive projection measurements to be preformed in a 3-dimensional Hilbert space.

Using this system, the authors demonstrate the convincing violation of the state dependent KCBS inequality in a generalised form taking into account the imperfect compatability of measurements. Although state-independent tests are more desirable (a point that could be mentioned in the paper), they are much harder to perform and remain a key next step in this research direction. Although recent loophole-free Bell-tests have also demonstrated quantum contextuality (enforced by the locality), the ability of contextuality to manifest itself in non-divisible systems means that this result is nonetheless important.

Response to the reviewer: We thank the reviewer for recognition of importance of our result.

The statistical analysis of the results performed is convincing and in line with the normal for tests of quantum nonlocality and contextuality, and is presented nicely in the paper and in more detail in the supplementary material (modulo a few formatting errors and the ungainly notation of writing "P - Value" instead of "p-value").

Response to the reviewer: We again thank for positive feedback on results and presentation. We proofread our manuscript and the supplementary information for formatting errors but did not find any. We use "P-value" notation following Ref. [1] of the Supplementary information.

One of the key difference in the analysis of noncontextuality inequalities for individual system vis à vis Bell-inequalities is the need to make the i.i.d. assumption that the authors discuss in the SM. Indeed, since quantum contextuality can be simulated with memory (and one cannot rule this out on grounds of locality for a single system), one cannot rule out such noncontextual HV theories. This point should perhaps be mentioned, and the paper "Memory Cost of Quantum Contextuality" (DOI: 10.1088/1367-2630/13/11/113011) is relevant to this point.

Response to the reviewer: We thank the reviewer for that interesting observation.

Changes to manuscript: We add the reference to the paper and modified the last paragraph of "Discussions".

Instead of: *"Recent theoretical work shows that extended noncontextuality inequalities[9,17] do not yet rule out all noncontextual HV models [18]. In the case of the finite-precision loophole [25,26], debate continues about whether this loophole can be closed in principle [27-29]. It remains an important open challenge to identify a clear, general prescription for how to implement a loophole-free test of noncontextuality."*

We say: *"One key point which differentiates our noncontextuality analysis for an indivisible system from a similar analysis for Bell inequalities with locality constraints, is that it is difficult to avoid the need for additional i.i.d. and no memory assumptions for measurements on a single system. Since quantum contextuality can be simulated by a classical system with memory [27], these loopholes will most likely remain for any Kochen-Specker tests without nonlocality. In another case, for the finite-precision loophole [28,29], debate continues about whether this loophole can be closed in principle[30-32]. It remains an important open*

challenge to identify a clear, general prescription for how to implement a noncontextuality test with minimal assumptions.”

Thus, for single systems, one cannot hope to close all possible loopholes as one might for Bell-inequalities, since this "memory" loophole remains, in addition to the finite-precision problem (although it is debatable that this is really a loophole). Given this, the paper seems to go as far as is possible for closing loopholes for single system Kochen-Specker tests, except for performing state-independent tests, and thus represents a key advance made possible by a very clever experimental setup.

Response to the reviewer: We thank the reviewer for the positive feedback. We are pleased to see how the reviewer summarized our work.

Overall, I think this paper is very suitable for publication in Nature Communications, with the minor points mentioned above taken into account.

Response to the reviewer: We again thank the reviewer for the positive assessment of our paper.

In addition, I have three very-minor, miscellaneous comments:

1) it is perhaps a stretch to call noncontextuality inequalities "recent" (paragraph 2), since they have been around for almost a decade.

Changes to manuscript: The word "recent" has been removed.

Instead of *"While the original theorem is difficult to test, the recent discovery of inequalities ..."*

We say: *"While the original theorem is difficult to test, the discovery of inequalities ..."*

2) (before Eq. (3)): I think it is a little misleading to attribute this "overt contextuality" to the Heisenberg Uncertainty Principle; it is simply due to noncommutativity of observables, and one can still have incompatible measurements with an underlying realist hidden variable theory.

Changes to manuscript: "... from the Heisenberg uncertainty principle ..." was removed.

Instead of *"... but this overt contextuality from the Heisenberg uncertainty principle does not reveal any further insight into the nature of reality."*

We say: *"...but this overt contextuality does not reveal any further insight into the nature of reality."*

3) In the discussion: the phrase "the last important loophole" is slightly misleading, since it would at face value imply that other loopholes are unimportant, which is not what is meant.

Changes to manuscript: "...the last important loophole" was removed.

Instead of *"The last important loophole, the compatibility loophole, was treated via an extended inequality."*

We say *"The compatibility loophole was treated via an extended inequality."*

Other changes to manuscript.

We found that Table I expectation values and P-values were evaluated for different datasets. To make all the data consistent we corrected P-values both in the main text and in the Supplementary Information which were evaluated for the same dataset. We also corrected typos in Table I: the numbers in round brackets for the $\langle A_j \rangle$ column represented the last significant figure not the standard deviation. It is now corrected.

Reviewer #1 (Remarks to the Author):

I was asked to review the revised manuscript "Contextuality without nonlocality in a superconducting quantum system." Most of my earlier criticism has been addressed. However, in my opinion, one major problem with the manuscript remains.

From my first report: "Unfortunately, the experiment shows a very clear influence of the measurement of the first observable on the second observable, cf. Table 1. This dependence is even context dependent, as can, e.g., be seen when comparing the observable 1 in the sequences (2,1) and (5,1) (the significance of this is more than 10 standard deviations)."

The authors give a very detailed answer to this concern, mainly confirming my observation and arguing that, on an absolute scale, the incompatibilities are small and also cannot be avoided. They also explain why this effect occurs. In addition the problem is treated by the specific error model chosen.

First, the experiment only excludes the particular error model. However, since the error model is not explained in the main text, the reader cannot understand even the main conclusion of the experiment.

Second, and more severe, there is a strong context dependence of the incompatibility. The effect is small on an absolute scale, but statistically highly significant. I understand that sometimes one cannot get rid of residual errors, in particular in a high precision experiment. But if these errors contradict the main claim of the experiment (i.e., a noncontextuality experiment has been performed), this should, at least, be emphasized in the text. Even better would of course be a quantitative analysis of the problem, cf., e.g., the rather similar case of the triple-slit experiment from the group of Weihs (Found. Phys. 42, 742 (2012), I guess), where such an analysis is provided.

Reviewer #2 (Remarks to the Author):

The authors demonstrate quantum contextuality without nonlocality with a superconducting qutrit system while some loopholes are closed. I am satisfied with the authors' response to my previous report, in particular they claim that there is no effect on the coherence between the first and the

second excited states at sweet spot which was experimentally proven using a series of measurements. I recommend publish this work in Nature Communications.

typos found:

The first sentence of the first paragraph, "...using underlying hidden variables (HVs) which determine, which determine the properties...". One of 'which determine' should be removed.

Reviewer #3 (Remarks to the Author):

All of the key points I raised in my previous review (modulo some new and not-new formatting errors, see below) appear to have been satisfactorily addressed, although it would have been nice to see the issue of state-(in)dependence mentioned at least briefly (as was raised indirectly by another referee also).

A couple of minor errors appear to have been introduced in the revision:

- "a such a" in the first paragraph, the first "a" of which should be deleted
- At the beginning of the second paragraph, rather than saying "the discovery of inequalities" it would be less ambiguous to say "the discovery of noncontextuality inequalities"
- A space between two sentences is missing in the discussion.

In the supplementary material, there are a few formatting problems, as I raised before:

- The term "P-value" is sometimes written with a simple hyphen, sometimes with a longer m-dash (perhaps it is written as $\$P-\$$ instead of $\$P\$$ and, in some displaymath, as $P -- \text{value}$ with extra spaces, which is very ungainly. This should be made consistent, preferable as " $\$P\$-value$ ".
- Equations 10-14 are indented with respect to Eqns. 15-18 when the = signs should all line up; the label for Eqn (17) hangs onto another line
- "Bell" should be capitalised in the bibliography.

REVIEWERS' COMMENTS:

Reviewer #1 (Remarks to the Author):

I was asked to review the revised manuscript "Contextuality without nonlocality in a superconducting quantum system." Most of my earlier criticism has been addressed. However, in my opinion, one major problem with the manuscript remains.

From my first report: "Unfortunately, the experiment shows a very clear influence of the measurement of the first observable on the second observable, cf. Table 1. This dependence is even context dependent, as can, e.g., be seen when comparing the observable 1 in the sequences (2,1) and (5,1) (the significance of this is more than 10 standard deviations)."

The authors give a very detailed answer to this concern, mainly confirming my observation and arguing that, on an absolute scale, the incompatibilities are small and also cannot be avoided. They also explain why this effect occurs. In addition the problem is treated by the specific error model chosen.

First, the experiment only excludes the particular error model. However, since the error model is not explained in the main text, the reader cannot understand even the main conclusion of the experiment.

Our response: The full formal definition of the model that we test is explained in the supplementary material. We do not define the full model formally in the main text, since the definition takes too much space page. However, we further would like to mention that the analysed model is as general as it gets for any testable i.i.d. model, where the requirement of bounded average incompatibility is needed for any realistic contextuality experiment, see Physical Review A 81.2 (2010): 022121. This was also pointed out by Reviewer #3 in his/her first review. We already included the statements about assumptions for i.i.d. model and no memory assumption in the main text (see page 4 right before Discussions and in Discussion section).

Second, and more sever, there is a strong context dependence of the incompatibility. The effect is small on an absolute scale, but statistically highly significant. I understand that sometimes one cannot get rid of residual errors, in particular in a high precision experiment. But if these errors contradict the main claim of the experiment (i.e., a noncontextuality experiment has been performed), this should, at least, be emphasized in the text. Even better would of course be a quantitative analysis of the problem, cf., e.g., the rather similar case of the triple-slit experiment from the group of Weihs (Found. Phys. 42, 742 (2012), I guess), where such an analysis is provided.

Our response: We do not share concerns of the reviewer for the above mentioned context dependence of the experimental data. We can repeat that such a context dependence is expected and even unavoidable for our experiment due to decoherence. However, it does not contradict to any our claims as our analysis does not rely on the assumption of the context-independence of the measured incompatibility.

We again kindly point the referee to the supplementary material, where it is spelled out there precisely what we test. In particular, we also test against models where the incompatibility is context-dependent. The only important requirement is that the average incompatibility is bounded. In Supplementary Note 1, section e we in fact also say how the incompatibility is estimated in a precise way. Both the statement of what we test as well as the analysis is done in a way, which, as we would respectfully like to point out, is more formal than the reference mentioned by the referee.

It is possible that the referee did not understand the argument in the supplementary material. While not necessary, we have undertaken efforts to explain ourselves better. In particular, we have rewritten section e in the Supplementary Note 1 in a possibly more didactic manner. Instead of just applying Bentkus' inequality (an established result in statistics which may not be so widely known), we first transform Bentkus' inequality into the form in which the more well-known Hoeffding' inequality is usually written. We then apply it in this familiar form, spelling out all details in how it is applied. We lose a bit in this conversion (the confidence is actually a bit worse) but since we remain in an excellent confidence regime we feel this possibly more didactic approach helps the reader to follow.

To summarize, we have taken great efforts to explain ourselves better, and we hope that a reader interested in the fine details of the analysis will be satisfied with the explanation in the supplementary material.

Reviewer #2 (Remarks to the Author):

The authors demonstrate quantum contextuality without nonlocality with a superconducting qutrit system while some loopholes are closed. I am satisfied with the authors' response to my previous report, in particular they claim that there is no affect on the coherence between the first and the second excited states at sweet spot which was experimentally proven using a series of measurements. I recommend publish this work in Nature Communications.

typos found:

The first sentence of the first paragraph, "...using underlying hidden variables (HVs) which determine, which determine the properties...". One of 'which determine' should be removed.

Our response: We thank the reviewer for spotting the typo. The typo is corrected.

Reviewer #3 (Remarks to the Author):

All of the key points I raised in my previous review (modulo some new and not-new formatting errors, see below) appear to have been satisfactorily addressed, although it would have been nice to see the issue of state-(in)dependence mentioned at least briefly (as was raised indirectly by another referee also).

Our response: We thank reviewer for his/her comment. In Discussion we added the sentence: "While we used the simpler state-dependent inequality for demonstration of contextual nature of the superconducting circuits, the state-independent test will be the straightforward extension of our experiments."

A couple of minor errors appear to have been introduced in the revision:

- *"a such a" in the first paragraph, the first "a" of which should be deleted.*
- *At the beginning of the second paragraph, rather than saying "the discovery of inequalities" it would be less ambiguous to say "the discovery of noncontextuality inequalities"*
- *A space between two sentences is missing in the discussion.*

Our response: We thank the reviewer for spotting the typos. We corrected them.

In the supplementary material, there are a few formatting problems, as I raised before:

- *The term "P-value" is sometimes written with a simple hyphen, sometimes with a longer m-dash (perhaps it is written as \$P-\$ instead of \$P\$ and, in some displaymath, as P -- value with extra spaces, which is very ungainly. This should be made consistent, preferable as "\$P\$-value".*
- *Equations 10-14 are indented with respect to Eqns. 15-18 when the = signs should all line up; the label for Eqn (17) hangs onto another line*
- *"Bell" should be capitalised in the bibliography.*

Our response: We thank the reviewer for spotting typos and formatting problems. The text is updated accordingly.